# Enhancing Methylene Blue Adsorption Performance of Ti_3_C_2_T_x_@Sodium Alginate Foam Through Pore Structure Regulation

**DOI:** 10.3390/nano14231925

**Published:** 2024-11-29

**Authors:** Yi Hu, Hongwei Wang, Xianliang Ren, Fang Wu, Gaobin Liu, Shufang Zhang, Haijun Luo, Liang Fang

**Affiliations:** 1Chongqing Key Laboratory of Interface Physics in Energy Conversion, College of Physics, Chongqing University, Chongqing 400044, Chinawufang@cqu.edu.cn (F.W.);; 2Key Laboratory on Optoelectronic Functional Materials, College of Physics and Electronic Engineering, Chongqing Normal University, Chongqing 401331, China; 3Center of Modern Physics, Institute for Smart City of Chongqing University in Liyang, Liyang 213300, China; 4Key Laboratory of Green and High-End Utilization of Salt Lake Resources, Qinghai Institute of Salt Lake, Chinese Academy of Sciences, Xining 810008, China

**Keywords:** Ti_3_C_2_T_x_, sodium alginate, methylene blue, adsorption, vacuum freeze-drying, honeycomb-like porous foam

## Abstract

Pore structural regulation is expected to be a facile way to enhance the adsorption performance of MXene. In this work, spherical foam composites consisting of Ti_3_C_2_T_x_ and sodium alginate (SA) were synthesized via a vacuum freeze-drying technique. By varying the solution volume of Ti_3_C_2_T_x_, four distinct Ti_3_C_2_T_x_@SA spherical foams with honeycomb-like and lamellar structures with a pore diameter in the range of 100–300 μm were fabricated. Their methylene blue (MB) adsorption performances were then systematically compared. The results revealed that the honeycomb-like porous-structured spherical foams have a significantly higher adsorption capacity than their lamellar counterparts. Notably, the Ti_3_C_2_T_x_@SA honeycomb-like porous foam exhibited a remarkable maximum adsorption capacity (*q_m_*) of 969 mg/g, positioning it at the forefront of MB adsorbent materials. Respective analysis of the adsorption kinetics, thermodynamics, and isotherm model indicated that this MB adsorption of Ti_3_C_2_T_x_@SA honeycomb-like porous foam is characterized to be a physical, endothermic, and monolayer adsorption. The Ti_3_C_2_T_x_@SA honeycomb-like porous foam also demonstrated excellent resistance to ion interference and good reusability, further attesting to its substantial potential for practical applications. X-ray photoelectron spectroscopy (XPS) analysis was employed to elucidate the adsorption mechanism, which was found to involve the synergistic effect of electrostatic adsorption and amidation reaction. This work not only offers new avenues for the development of high-performance adsorption materials but also provides crucial insights into the structural design and performance optimization of porous materials.

## 1. Introduction

Water pollution, particularly from organic dyes originating in industrial effluents, poses a significant threat to global aquatic ecosystems. Methylene blue (MB, C_16_H_18_N_3_ClS), a quintessential cationic dye with planar structure, has become a pivotal focus in environmental science research due to its widespread applications in the textile, papermaking, and pharmaceutical industries [1]. The molecular architecture of MB is distinguished by the incorporation of three nitrogen atoms covalently bonded to carbon atoms within both the benzene ring and the heterocyclic system. Additionally, the molecule features a methylene group appended to a carbon atom in the heterocyclic ring, along with two dimethylamino groups and a sulfur atom, collectively forming the phenothiazine ring structure [2]. Notably, MB exhibits remarkable stability under visible light irradiation compared with other organic dyes and is resistant to degradation solely through photolysis or catalysis [3]. Its presence in water bodies not only leads to visual pollution but also poses serious risks to both ecosystems and human health, given its potential carcinogenicity and toxicity [4]. Consequently, developing efficient and cost-effective technologies for MB removal is essential for safeguarding water resources and public health.

Among the array of water treatment technologies, adsorption stands out for its simplicity, cost-effectiveness, and high efficiency [5,6]. Recently, two-dimensional (2D) materials have become a focal point in adsorption research due to their unique physicochemical properties, including high specific surface area, exceptional mechanical strength, and excellent chemical stability [7,8]. Notably, Ti_3_C_2_T_x_ (MXene), a representative 2D transition metal carbide and nitride, has shown promise in water treatment applications due to its remarkable metallic conductivity and hydrophilicity [9]. However, the standalone use of MXene faces challenges such as a propensity for agglomeration and difficulty in recovery, which can limit its practical application efficiency.

To address these limitations, researchers have investigated the composite of MXene with other materials. Sodium alginate (SA), a natural polysaccharide derived from brown algae, is an ideal candidate for combination with MXene due to its biocompatibility, biodegradability, and environmental benignity [10]. SA and its composites, as adsorbents, can maintain their adsorptive properties post-desorption, making them promising candidates among biopolymer materials. The current state of research on SA composites in adsorption indicates their superior performance in removing a variety of pollutants. Hao’s research team has reported that graphene oxide-montmorillonite/SA aerogel beads exhibit remarkable adsorption capacity and selectivity for MB in wastewater, with a *q_m_* reaching 150 mg/g [11]. Wang et al. have introduced nickel alginate/activated carbon aerogel and nickel alginate/graphene oxide aerogel, which achieved *q_m_* values of 465 mg/g and 537 mg/g for MB, respectively [12]. Gong et al. have investigated the adsorption of crystal violet dye by sugarcane bagasse-bentonite/SA composite aerogel, attaining a *q_m_* of 839 mg/g [13]. The Marzban team has successfully encapsulated kaolin powder within SA, resulting in a 99.56% removal efficiency and a *q_m_* of 188.7 mg/g for MB. This represents a more than fourfold increase in comparison to the use of kaolin alone [14]. The preparation of porous materials through the combination of SA and MB adsorbent materials constitutes an effective strategy for augmenting adsorption capacity. Nonetheless, the investigation on the influences of the pore structure of these porous materials on the adsorption performance is still relatively rare.

This study is aimed at obtaining MXene-based porous materials with high adsorption capacity, easy recoverability, and good reusability. Therefore, the composite of Ti_3_C_2_T_x_ and SA was designed and prepared by utilizing a straightforward vacuum freeze-drying technique. The influence of the preparation methodology on the structural and MB adsorption properties of the Ti_3_C_2_T_x_@SA porous materials was studied. Then, the adsorption process was in-depth analyzed and the underlying mechanisms of MB onto the composites was elucidated. It is envisioned that this work will provide some new idea to exploit novel MXene-based high-performance adsorbent materials and offer fresh perspectives for the treatment of MB-containing wastewater.

## 2. Experimental and Methods

### 2.1. Material

All materials were used directly upon purchase without any special treatment. Commercial Ti_3_AlC_2_ (400 mesh) was purchased from Xinxi Science and Technology, Foshan, China. Sodium alginate (SA, AR) was purchased from Shanghai Macklin Co., Ltd., Shanghai, China. Lithium fluoride (LiF, 99%), hydrochloric acid (HCl, 9 M), sodium hydroxide (NaOH, 5 M), and methylene blue (99.9%) were all purchased from Shanghai Aladdin Co., Ltd., Shanghai, China. Calcium chloride (CaCl_2_, AR) was obtained from Chongqing Chuandong Chemical Co., Ltd., Chongqing, China.

### 2.2. Preparation of Ti_3_C_2_T_x_

The synthesis of Ti_3_C_2_T_x_ is schematically illustrated in Figure 1. LiF and 9 M HCl were combined in a 1.6 g-to-20 mL ratio and homogeneously mixed under magnetic stirring to prepare the etching solution. Subsequently, 1 g of the Ti_3_AlC_2_ precursor was introduced into the etching solution and subjected to magnetic stirring at 500 rpm at a temperature of 45 °C for a duration of 48 h to facilitate etching. Upon completion of the etching process, the etched material was purified by successive centrifugal washes with deionized water, continuing until the pH of the supernatant reached approximately 6, thereby removing any residual ionic impurities. Ti_3_C_2_T_x_ was then subjected to ultrasonication in cold water for a period of 2 h to assist in the exfoliation process and to obtain a few-layered colloidal solution of Ti_3_C_2_T_x_. Post-ultrasonication, the solution was centrifuged at 3500 rpm for 30 min, after which the supernatant was carefully collected to isolate the few-layered Ti_3_C_2_T_x_ solution. The density of the material was subsequently determined following freeze-drying.

### 2.3. Preparation of Ti_3_C_2_T_x_@SA Foam

The fabrication process of Ti_3_C_2_T_x_@SA foam spheres is shown in Figure 1. One g SA powder was individually dissolved in 10, 20, 40, and 60 mL volumes of a Ti_3_C_2_T_x_ solution with a concentration of 4.85 mg/L. Following 30 min of intense stirring, a viscous colloidal intermediate was obtained. The intermediate was incrementally introduced into a 2% calcium chloride (CaCl_2_) solution through a peristaltic pump, facilitating crosslinking for a duration of 6 h. The materials were then subjected to vacuum freeze-drying; then, the Ti_3_C_2_T_x_@SA foam spheres were yielded. These samples corresponding to the used volumes of Ti_3_C_2_T_x_ solution (10, 20, 40, and 60 mL) were labeled as 1-Ti_3_C_2_T_x_@SA, 2-Ti_3_C_2_T_x_@SA, 4-Ti_3_C_2_T_x_@SA, and 6-Ti_3_C_2_T_x_@SA, respectively.

### 2.4. Characterization

The functional groups were characterized utilizing a Nicolet iS50 Fourier-transform infrared spectrometer (FTIR), Thermo Fisher Scientific, Waltham, MA, USA. The crystallographic analysis of Ti_3_C_2_T_x_ was conducted with the aid of a Philips X’Pert Powder X diffractometer (XRD), Spectris plc, London, UK. Morphological studies were carried out with a JEOL JSM7800F scanning electron microscope (SEM), JEOL Ltd., Tokyo, Japan. XPS data were acquired on a Thermo Scientific ESCALAB 250Xi system, Thermo Fisher Scientific, Waltham, MA, USA. Quantitative ion concentration measurements were made possible by a Shimadzu UV-3600 spectrophotometer, Shimadzu Corporation, Kyoto, Japan.

### 2.5. Batch Adsorption Experiments

During the adsorption process, a volume of 100 mL of MB solution was utilized, into which 20 mg of Ti_3_C_2_T_x_@SA foam were introduced and magnetically stirred at a speed of 500 rpm. All adsorption experiments, with the exception of those related to thermodynamic models, were conducted at room temperature, specifically at 25 °C. All adsorption experiments were repeated three times.

The adsorption capacity refers to the mass or amount of solute adsorbed per unit mass of adsorbent. This is a crucial parameter for evaluating the performance of an adsorbent, and it is calculated using the following formulas [15,16]:(1)$$q_{t}=\frac{(C_{0}-C_{t})V}{m}$$
(2)$$q_{e}=\frac{(C_{0}-C_{e})V}{m}$$

The removal efficiency pertains to the proportion of a specific contaminant that is eliminated from a solution by adsorption. The formula for calculating the removal rate is as follows [17]:(3)$$R_{e}\%=\frac{C_{0}-C_{e}}{C_{0}}\times 100$$
where *q_t_* (mg/g) denotes the adsorption capacity at time *t*, *q_e_* (mg/g) represents the adsorption capacity at equilibrium, *r* signifies the removal efficiency, *C*_0_ (mg/g) is the initial concentration of the solution, *C_t_* (mg/g) indicates the concentration of the solution at time *t*, *C_e_* (mg/g) reflects the concentration of the solution at equilibrium, *V* (L) stands for the volume of the solution, and *m* (g) is the mass of the adsorbent employed.

The impact of pH on adsorption capacity is multifaceted, encompassing a variety of intricate interactions and mechanisms. To systematically investigate this effect, a series of MB solutions were meticulously prepared with precise pH values of 3.18, 5.12, 7.14, 8.98, 10.01, and 10.97, while all other experimental parameters (concentration, solution volume, adsorbent dosage, temperature, adsorption duration, and stirring speed) were rigorously controlled to ensure consistency. The adsorption capacity and removal efficiency of contaminants were then carefully calculated at adsorption equilibrium for each pH level. Through a comprehensive comparison of the adsorption performance across the tested pH range, the optimal pH condition for maximum adsorption was determined.

Adsorption kinetics experiments are conducted to explore the time-varying interactions between adsorbents and adsorbates during the adsorption process, serving as a means to investigate the factors governing the adsorption rate [18]. To ensure the accuracy of the results, it is imperative to maintain stable conditions, including temperature and stirring speed, throughout the experiments. Samples were collected at predefined time points (1, 3, 5, 7, 15, 21, 28, and 36 h). The concentration of MB in the solution after adsorption was measured, and the corresponding adsorption capacity was calculated. Subsequently, a curve was plotted to illustrate the change in adsorption capacity over time. By fitting the data to adsorption kinetics models, such as the pseudo-first-order and pseudo-second-order kinetic models, relevant kinetic parameters were determined. Furthermore, the intraparticle diffusion model was employed to study the diffusion process of adsorbates within the adsorbent. The fitting formula for the pseudo-first-order kinetic model is as follows [19]:(4)$$q_{t}{=q}_{e}\;(1-{\;e}^{{-K}_{1}t})$$

The pseudo-second-order kinetic model is as follows [20]:(5)$$\frac{t}{q_{t}}=\frac{1}{k_{2}q_{e}^{2}}+\frac{t}{q_{e}}$$

The intraparticle diffusion model is as follows [21]:(6)$$q_{t}=k_{i}t^{\frac{1}{2}}+C_{i}$$
where *k*_1_ and *k*_2_ are the constants for the corresponding kinetic models, and *k_i_* represents the coefficient for stage *i* in the intraparticle diffusion model. *q_t_* (mg/g) is the adsorption capacity at time *t*, *q_e_* (mg/g) is the adsorption capacity at equilibrium, *t* is the adsorption time, and *C_i_* (mg/g) is the solution concentration at time *i*.

The adsorption isotherm model characterizes the relationship curve describing the equilibrium concentrations of solute molecules at the interface between two phases under constant temperature conditions [22]. With predefined parameters, including temperature, pH, adsorbent dosage, and stirring speed, the initial concentration of MB was systematically varied (set at 100, 150, 200, 250, and 300 mg/L). Upon reaching adsorption equilibrium, samples were collected to analyze the correlation between adsorption capacity and initial MB concentration. Subsequently, the Langmuir and Freundlich models were utilized to fit the experimental data, with a focus on enhancing the understanding of the adsorption process. The equations for the Langmuir and Freundlich isotherm models are presented as follows:

Langmuir isotherm model [23]:(7)$$q_{e}=\frac{q_{m}K_{L}C_{e}}{{1+K}_{L}C_{e}}$$

Freundlich isotherm model [24]:(8)$$q_{e}{=K}_{F}C_{e}^{n}$$
where *C_e_* (mg/L) denotes the equilibrium concentration of MB, *q_e_* (mg/g) represents the equilibrium adsorption capacity, and *q_m_* (mg/g) is the theoretical maximum adsorption capacity in the Langmuir model. The parameter *n* indicates the adsorption intensity in the Freundlich model, while *K_L_* and *K_F_* are the equilibrium adsorption constants for the Langmuir and Freundlich models, respectively.

When investigating the impact of temperature on adsorption performance, 20 mg of foam were used to adsorb 100 mL of a 150 mg/L MB solution at temperatures of 298, 308, and 318 K, respectively. The specific calculation formulas are as follows [25,26]:(9)$$K_{d}=\frac{(C_{0}-C_{e})V}{mC_{e}}$$
(10)$$\ln K_{d}=\frac{\Delta S}{R}-\frac{\Delta H}{RT}$$
(11)ΔG=ΔH−T×ΔS
where *K_d_* denotes the distribution coefficient (mL g^−1^), *T* represents the absolute temperature (K), and *R* is the ideal gas constant (8.314 J mol^−1^ K^−1^). Additionally, Δ*H* signifies the enthalpy change, Δ*S* the entropy change, and Δ*G* the Gibbs free energy change.

### 2.6. Selective Adsorption and Recyclability Experiments

The impact of coexisting ions on the selective adsorption of methylene blue (MB) by Ti_3_C_2_T_x_@SA foam was investigated. In this study, the concentrations of coexisting ions (KCl, NaNO_3_, MgSO_4_) and MB were both set at 0.3 g/L. At pH 10, 20 mg of 1-Ti_3_C_2_T_x_@SA foam were used to adsorb 100 mL of a 300 mg/L pure MB solution and an MB solution containing the coexisting ions, respectively, with the adsorption capacities being compared.

To assess the reusability of the foam, desorption and re-adsorption experiments were conducted. The foam, after adsorbing 100 mL of 300 mg/L MB, was immersed in ethanol, and the pH was adjusted to 2 using HCl. It was then ultrasonically cleaned until the solution became colorless and transparent. Subsequently, the treated foam was again utilized to adsorb 100 mL of 300 mg/L MB solution at pH 10. This process was repeated multiple times, and changes in adsorption capacity were compared to evaluate the reusable performance of the 1-Ti_3_C_2_T_x_@SA foam.

## 3. Result and Discussion

### 3.1. Structure and Morphology

The XRD patterns of Ti_3_AlC_2_, Ti_3_C_2_T_x_, and 1-Ti_3_C_2_T_x_@SA are presented in Figure 2A. It shows that the precursor Ti_3_AlC_2_ exhibits three distinct XRD diffraction peaks located at 9.6° (002), 19.1° (004), and 38.8° (104). However, following etching treatment, the (002) plane diffraction peak of Ti_3_C_2_T_x_ is shifted toward a smaller angle at 7.51° from the original 9.62°. Concurrently, in the XRD pattern of Ti_3_C_2_T_x_, the diffraction peak originally corresponding to the (104) plane of Ti_3_AlC_2_ at 38.8° almost completely disappears. This shift indicates that the Al layer in the precursor has been successfully etched away, confirming the successful preparation of Ti_3_C_2_T_x_ [27]. The XRD pattern of the 1-Ti_3_C_2_T_x_@SA foam reveals a pronounced (002) diffraction peak, which has shifted to a lower angle of 5.58°. This shift suggests a significant expansion of the interlayer spacing within the Ti_3_C_2_T_x_ lattice upon incorporation into the foam matrix [28].

The FTIR spectra of pure SA and SA foam composited with Ti_3_C_2_T_x_ are displayed in Figure 2B. Two distinct infrared absorption peaks are observed at 3428 and 1656 cm^−1^, which correspond to the stretching vibrations of –OH and –COOH, respectively [29,30]. Given that the FTIR spectrum of Ti_3_C_2_T_x_ also displays a stretching vibration for –OH, no significant changes are evident in the FTIR spectrum of the composite sample.

As depicted in Figure 3A, Ti_3_AlC_2_ is observed as a bulk material with dimensions of 22 × 13 μm, characterized by tightly packed layers and a stable structure [31]. As evidenced in Figure 3B, following the etching process, two-dimensional flake material—few-layer Ti_3_C_2_T_x_—with dimensions of approximately 16 × 12 μm was successfully synthesized.

The SEM morphology of Ti_3_C_2_T_x_@SA foam spheres prepared with various volumes of Ti_3_C_2_T_x_ solution are demonstrated in Figure 3D–O. These foam spheres possess a diameter of approximately 3 mm, featuring wrinkled surfaces devoid of noticeable pores. Upon bisecting a 1-Ti_3_C_2_T_x_@SA foam sphere, the cross-sectional SEM image reveals a honeycomb-like porous internal structure [32]. The diameters of the pores primarily fall within the range of 100–300 μm. Notably, the sphere fabricated using 10 mL of Ti_3_C_2_T_x_ solution exhibits the most uniform and densely packed internal pore distribution, closely resembling a honeycomb structure [33]. Conversely, as the volume of Ti_3_C_2_T_x_ solution used in the preparation of the foam spheres increases, the attainment of a uniform internal structure becomes increasingly challenging, with a heightened risk of collapse and stacking. The internal structure of 2-Ti_3_C_2_T_x_@SA resembles a petal-like, well-defined layered structure. Both 4-Ti_3_C_2_T_x_@SA and 6-Ti_3_C_2_T_x_@SA also exhibit layered structures internally, but with noticeable stacking occurring.

### 3.2. Comparison of MB Adsorption Capacity of Four Ti_3_C_2_T_x_@SA Foam Samples

Figure 4 illustrates the equilibrium adsorption capacities of the four types of Ti_3_C_2_T_x_@SA foams prepared with various solution volumes for the adsorption of 100 mg/L MB solution at pH 6.5. It was observed that the adsorption capacity of the foams decreased with an increase in the volume of Ti_3_C_2_T_x_ used during preparation. Notably, the foam prepared with 10 mL of Ti_3_C_2_T_x_ solution exhibited the highest adsorption capacity, reaching 405 mg/g. Therefore, the 1-Ti_3_C_2_T_x_@SA foam samples were selected for subsequent experiments to further investigate their adsorption performance towards MB. Pores with regular honeycomb shapes are more conducive to the uniform transport of the solution, while pores with complex shapes may result in tortuous and irregular transport paths, thereby increasing the difficulty of transport.

### 3.3. Effect of pH on Adsorption Capacity of 1-Ti_3_C_2_T_x_@SA Foam

The pH environment is a direct determinant of adsorption performance. In an effort to identify the most favorable conditions for the application of 1-Ti_3_C_2_T_x_@SA foam, the equilibrium adsorption capacity was evaluated at different pH levels for 20 mg of the foam on 100 mL of a 100 mg/L MB solution, as illustrated in Figure 5. Across the pH range from 3 to 10, the adsorption capacity of 1-Ti_3_C_2_T_x_@SA foam incrementally rose with increasing pH, peaking at 420 mg/g at pH 10. Interestingly, a further increase in pH to 11 resulted in a decrease in adsorption capacity. The adsorptive capacity of 1-Ti_3_C_2_T_x_@SA foam is notably robust within the pH spectrum of 5 to 10, demonstrating its effectiveness across a broad range of acidic to neutral conditions.

The impact of pH on adsorption performance is intricate, encompassing the surface charge dynamics of the adsorbent and the protonation state of MB [34]. Figure 5B illustrates that Ti_3_C_2_T_x_ exhibits an isoelectric point at pH 2.64, below which the surface of Ti_3_C_2_T_x_ is positively charged and above which it becomes negatively charged, with the surface negative charge intensifying as pH increases. In tandem, the electronegativity of the SA surface also escalates with rising pH levels. Consequently, the electronegativity of 1-Ti_3_C_2_T_x_@SA foam augments with increasing pH, enhancing its electrostatic attraction to the positively charged MB. Conversely, in alkaline conditions, the elevation of pH leads to the combination of MB with OH^−^, diminishing its positive charge and consequently weakening the electrostatic adsorption, which in turn reduces the adsorption capacity of 1-Ti_3_C_2_T_x_@SA foam [35]. The optimal pH condition for the application of 1-Ti_3_C_2_T_x_@SA foam is 10, so subsequent adsorption experiments are all conducted under this condition and will not be further elaborated.

### 3.4. Adsorption Kinetics and Isotherm Models

Figure 6A depicts the time-dependent adsorption capacity of 20 mg 1-Ti_3_C_2_T_x_@SA foam for MB at concentrations of 100, 150, and 200 mg/L. Complete adsorption equilibrium was achieved within 36 h for the 1-Ti_3_C_2_T_x_@SA foam. The fitting results of the pseudo-first-order and pseudo-second-order kinetic models indicated a higher degree of conformity between the pseudo-first-order kinetic model and the experimental data. A detailed comparison of the fitting parameters, as listed in Table 1, clearly demonstrates that the theoretical equilibrium adsorption capacity derived from the PFO kinetic model aligns more closely with the experimental results. This further substantiates the suitability of the PFO kinetic model for describing the adsorption process of MB onto 1-Ti_3_C_2_T_x_@SA foam. The adherence of the adsorption process to the PFO kinetic model suggests that the adsorption is likely diffusion-controlled and represents a reversible physical adsorption process [36].

The fitting results of Weber’s intraparticle diffusion model are presented in Figure 6B. The initial stage, occurring within the first 16 h of adsorption, is characterized by rapid diffusion of MB to the outer surface of the 1-Ti_3_C_2_T_x_@SA foam and subsequent surface adsorption. The second stage, from 16 to 32 h, is marked by saturation of the outer surface with adsorbed MB and penetration of MB into the interlayer spaces of the foam, where it undergoes diffusion and subsequent adsorption. The third stage is reached after 32 h, where a dynamic equilibrium between adsorption and desorption is established, and at this point, *q_t_* no longer increases. The fitting line of the intraparticle diffusion model does not intersect the origin, indicating that the adsorption process is governed by both external and internal particle diffusion. The steeper slope of the first stage compared to the second stage suggests a faster adsorption rate and higher adsorption capacity in the initial phase, indicating that external diffusion predominantly influences the adsorption rate [37]. A detailed comparison of the coefficients at each stage is presented in Table 2.

Figure 7A depicts the relationship between the equilibrium adsorption capacity and the initial MB concentration for 20 mg of 1-Ti_3_C_2_T_x_@SA. As the MB concentration gradually rises from 100 mg/L to 300 mg/L, the equilibrium adsorption capacity of 1-Ti_3_C_2_T_x_@SA increases progressively, attaining a *q_m_* of 969 mg/g. The fitting results using the Langmuir and Freundlich adsorption isotherm models, presented in Figure 7B, reveal that the Langmuir model exhibits a significantly higher correlation coefficient (R^2^ = 0.9829) than the Freundlich isotherm model (R^2^ = 0.9622), indicating a superior fit to the experimental data. This suggests that the adsorption sites on the adsorbent surface are uniformly distributed, with MB forming a monolayer adsorption on the 1-Ti_3_C_2_T_x_@SA surface [38]. A comparison of the actual measured maximum adsorption capacity of 1-Ti_3_C_2_T_x_@SA for MB with existing reports (Table 3) clearly showcases that the adsorption capacity of 1-Ti_3_C_2_T_x_@SA is among the highest reported.

### 3.5. Adsorption Thermodynamics

The effect of temperature on the adsorption capacity of 1-Ti_3_C_2_T_x_@SA for MB was examined at temperatures of 298 K, 308 K, and 318 K. The experimental data show a consistent upward trend in *q_e_* with increasing temperature (as depicted in Figure 8A), providing strong evidence that the adsorption process is endothermic. Upon further analysis of Figure 8B, a significant linear relationship was observed between lnKd and the reciprocal of temperature (1/T). The calculated values for ΔH, ΔS, and ΔG are summarized in Table 4. Specifically, the positive ΔH value reaffirms the endothermic character of the adsorption process. Moreover, it was noted that ΔS is also positive, indicating an increase in the system’s entropy during adsorption. The negative ΔG value suggests that, from a thermodynamic standpoint, the adsorption reaction is spontaneous. As the temperature rises, the negativity of ΔG becomes more pronounced, indicating an increased spontaneity of the adsorption reaction with elevated temperature.

### 3.6. Selective Adsorption Experiment and Recyclability

The present study examines the influence of interfering ions on the adsorption capabilities of 1-Ti_3_C_2_T_x_@SA, with a detailed analysis of variations in adsorption capacity in the presence of different ions, as depicted in Figure 9. The findings reveal that the presence of cations markedly impacts adsorption capacity: Specifically, the adsorption capacity diminishes by 4.95%, 6.81%, and 10.83% in the presence of K^+^, Mg^2+^, and Na^+^ ions, respectively, relative to a control group devoid of ionic interference. This decrement can be attributed to the direct competition between these cations and the target pollutant, MB, for electrostatic adsorption sites on the adsorbent surface, consequently diminishing adsorption capacity.

The impact of anions, including Cl^−^, SO_4_^2−^, and NO_3_^−^, on the adsorption capacity of 1-Ti_3_C_2_T_x_@SA for MB is comparatively modest, resulting in reductions of 0.82%, 1.65%, and 2.57%, respectively. This finding indicates that, owing to their negative charge, these anions do not engage in direct competition with the positively charged MB for adsorption sites, thereby having a minimal effect on the adsorption process.

The reusability of 1-Ti_3_C_2_T_x_@SA foam was rigorously assessed through five cycles of adsorption and desorption. As depicted in Figure 9B, despite a minor decrease in adsorption performance with each subsequent desorption cycle, the foam retains an adsorption capacity of 785 mg/g after the fifth cycle, which corresponds to 81% of its initial capacity. This finding demonstrates that 1-Ti_3_C_2_T_x_@SA foam sustains high adsorption efficiency even after repeated use.

Considering that the adsorption mechanism of 1-Ti_3_C_2_T_x_@SA foam is predominantly driven by electrostatic interactions, the desorption process is notably facile. This attribute not only validates the foam’s reusability but also suggests that, in practical applications, 1-Ti_3_C_2_T_x_@SA foam can be efficiently regenerated via straightforward desorption steps, thus minimizing operational costs and enhancing economic viability.

### 3.7. Mechanism of MB Adsorption by 1-Ti_3_C_2_T_x_@SA Porous Composite

To gain a deeper understanding of the mechanism underlying the removal of MB by 1-Ti_3_C_2_T_x_@SA, changes in chemical bonds during the adsorption process were analyzed through XPS by examining shifts in binding energies before and after element adsorption. Figure 10A presents the results of the XPS full spectrum scan. Prior to adsorption, only Ti, C, O, and F elements are present in 1-Ti_3_C_2_T_x_@SA. Following adsorption, the emergence of an N 1s peak provides direct evidence of successful MB adsorption. Further analysis of the high-resolution XPS spectrum of C 1s revealed no significant changes before and after adsorption (Figure 10B), indicating that C atoms do not undergo chemical changes during the adsorption process. After adsorption, the Ti 2p spectrum exhibits a Ti-N peak at 457.34 eV (Figure 10C), suggesting that during adsorption, N atoms, with their strong electronegativity and lone-pair electrons, form stable covalent bonds with the empty orbitals of Ti atoms. The high-resolution N 1s spectrum also shows no N peak before adsorption, but after adsorption, a C–N peak at 398.32 eV attributed to MB appears (Figure 10D), confirming effective adsorption of MB onto 1-Ti_3_C_2_T_x_@SA.

Based on a comprehensive analysis of zeta potential and XPS results, the adsorption mechanism is proposed in Figure 11. After the compounding of Ti_3_C_2_T_x_ with SA, the spherical foam becomes negatively charged in an alkaline environment, enabling electrostatic adsorption with positively charged MB and enhancing the physical adsorption of SA. The carboxyl groups in 1-Ti_3_C_2_T_x_@SA can undergo dehydration condensation with the amino groups in MB to form amide bonds, releasing a molecule of water. The reaction can be represented as follows [46]:R−COOH + NH_2_−R’ → R−CONH−R’ + H_2_O(12)

Furthermore, XPS results indicate that during the adsorption process, Ti atoms from Ti_3_C_2_T_x_ also form covalent bonds with N atoms from MB, contributing to the overall enhancement of adsorption.

## 4. Conclusions

In this work, four distinct Ti_3_C_2_T_x_@SA spherical foams, featuring internal honeycomb-like and lamellar structures, were successfully synthesized through a vacuum freeze-drying technique. A comprehensive assessment was subsequently conducted to compare their adsorption performances. The findings revealed that the structured honeycomb-like variant exhibited superior adsorption capabilities. Further investigation was then carried out to elucidate the adsorption behavior of MB by the 1-Ti_3_C_2_T_x_@SA spherical foam with a honeycomb-like structure. Over a broad pH range of 5 to 10, 1-Ti_3_C_2_T_x_@SA demonstrated remarkable MB adsorption capacity, peaking at a pH of 10 with a maximum adsorption capacity of 969 mg/g, which is among the highest reported for MB adsorbents. The adsorption process was characterized as spontaneous, endothermic, and consistent with a monolayer adsorption model, primarily driven by the synergistic effects of electrostatic adsorption and amidation reaction. Additionally, 1-Ti_3_C_2_T_x_@SA exhibited exceptional resistance to ion interference and robust reusability, collectively suggesting its promising potential for diverse applications and substantial practical value in the adsorption field.

## Figures and Tables

**Figure 1 nanomaterials-14-01925-f001:**
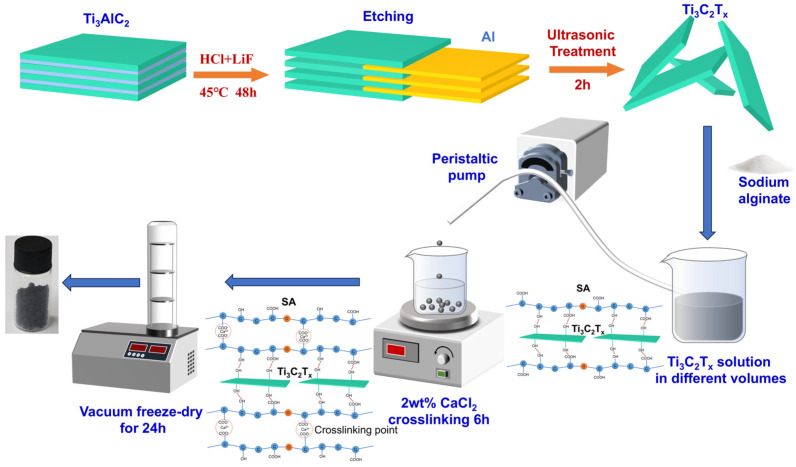
Schematic illustration of Ti_3_C_2_T_x_ synthesis and the fabrication of Ti_3_C_2_T_x_@SA foam spheres.

**Figure 2 nanomaterials-14-01925-f002:**
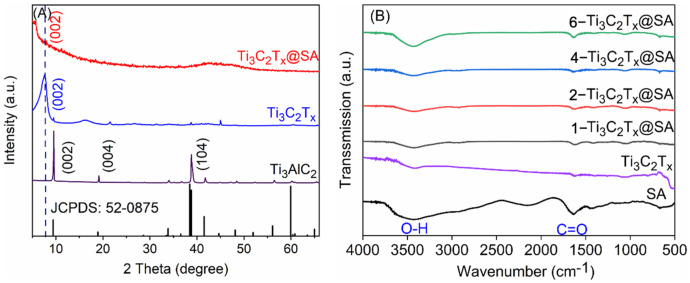
(**A**) The XRD patterns of Ti_3_AlC_2_, Ti_3_C_2_T_x_, and 1-Ti_3_C_2_T_x_@SA; (**B**) the FTIR spectra of SA, Ti_3_C_2_T_x_, and four Ti_3_C_2_T_x_@SA samples.

**Figure 3 nanomaterials-14-01925-f003:**
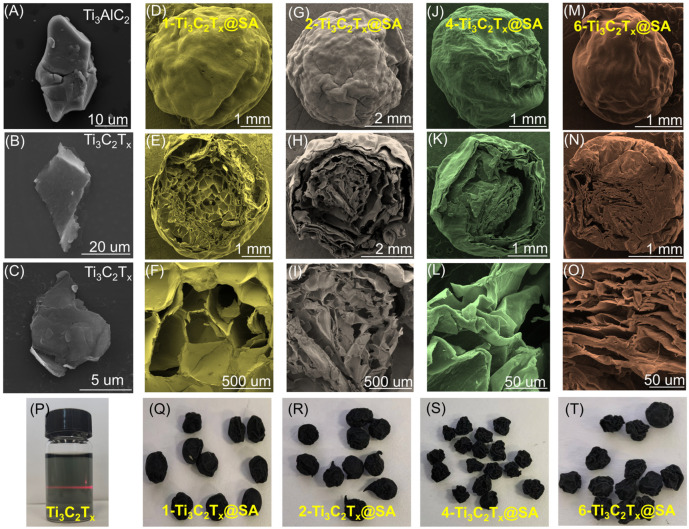
SEM morphology of (**A**) Ti_3_AlC_2_, (**B**,**C**) Ti_3_C_2_T_x_, (**D**–**F**) 1-Ti_3_C_2_T_x_@SA, (**G**–**I**) 2-Ti_3_C_2_T_x_@SA, (**J**–**L**) 4-Ti_3_C_2_T_x_@SA, (**M**–**O**) 6-Ti_3_C_2_T_x_@SA, and (**P**–**T**) physical photograph.

**Figure 4 nanomaterials-14-01925-f004:**
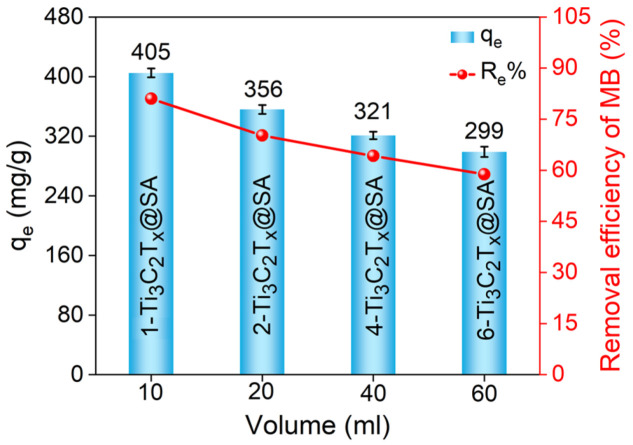
Comparison of adsorption performance of samples with different internal structures prepared by adjusting the volume of Ti_3_C_2_T_x_ solution during compounding.

**Figure 5 nanomaterials-14-01925-f005:**
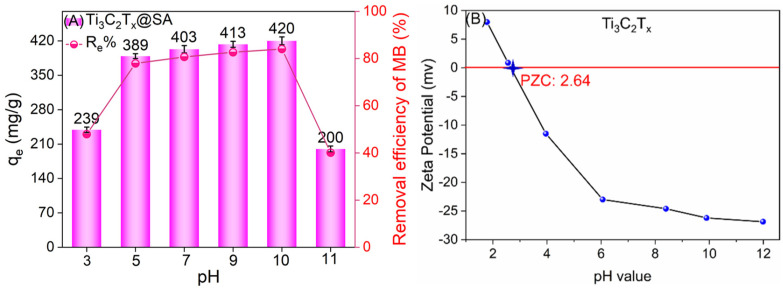
(**A**) Impact of pH on the adsorption capacity of 1-Ti_3_C_2_T_x_@SA foam; (**B**) relationship between the zeta potential of Ti_3_C_2_T_x_ and pH.

**Figure 6 nanomaterials-14-01925-f006:**
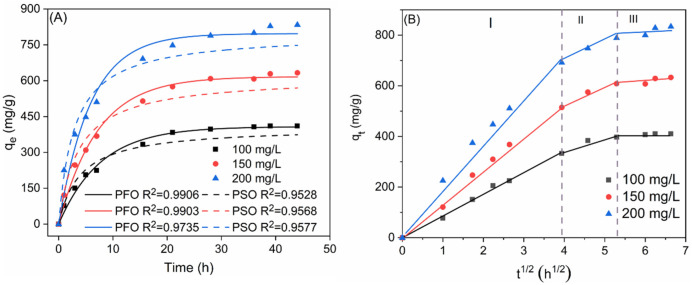
(**A**) Time-dependent MB adsorption capacity of 20 mg 1-Ti_3_C_2_T_x_@SA foam at pH = 10; (**B**) Weber–Morris’s intraparticle diffusion model.

**Figure 7 nanomaterials-14-01925-f007:**
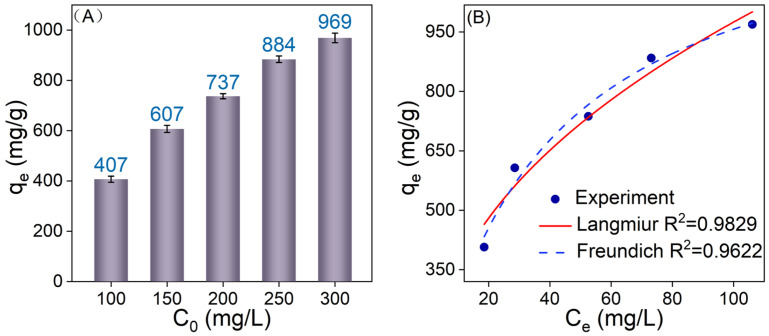
(**A**) Variation in adsorption capacity of 1−Ti_3_C_2_T_x_@SA foam with initial MB concentration; (**B**) fitting results of Langmuir and Freundlich adsorption isotherms.

**Figure 8 nanomaterials-14-01925-f008:**
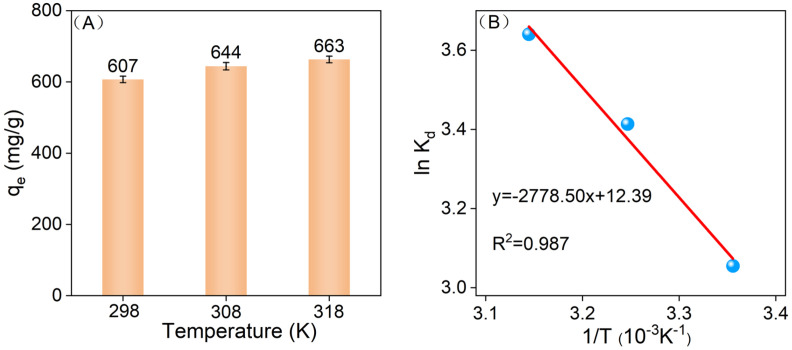
(**A**) *q_e_* of 20 mg 1-Ti_3_C_2_T_x_@SA on 150 mg/L MB at 298, 308, 318 K; (**B**) Van der Waals equation fitting.

**Figure 9 nanomaterials-14-01925-f009:**
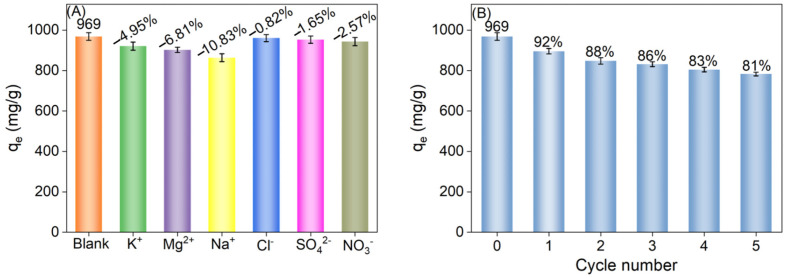
(**A**) The impact of Na^+^, Mg^2+^, K^+^, SO_4_^2−^, Cl^−^, and NO_3_^−^ on the q_m_ of 20 mg 1-Ti_3_C_2_T_x_@SA to 100 mL 300 mg L^−1^ MB; (**B**) adsorption capacity over five consecutive cycles of adsorption and desorption.

**Figure 10 nanomaterials-14-01925-f010:**
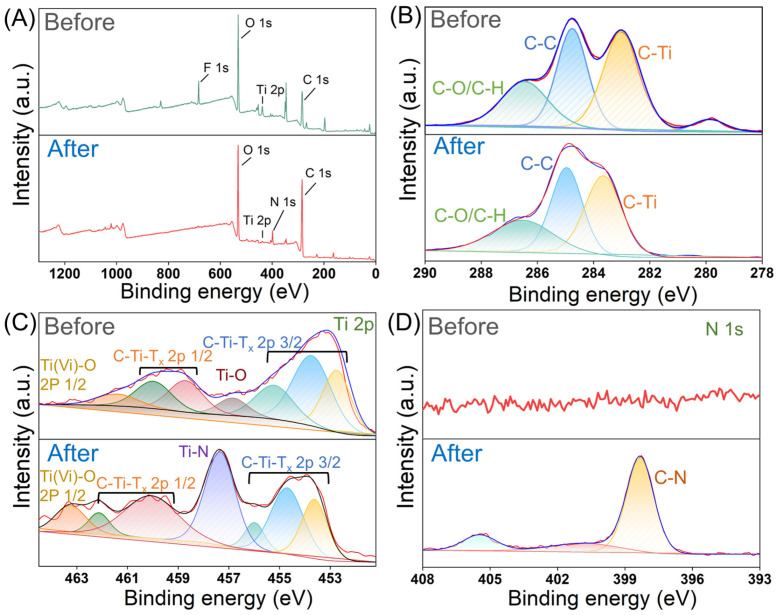
XPS spectra of 1-Ti_3_C_2_T_x_@SA before and after MB removal. (**A**) Survey scan and high-resolution spectra of C 1s (**B**), of Ti 2p (**C**), and of N 1s (**D**).

**Figure 11 nanomaterials-14-01925-f011:**
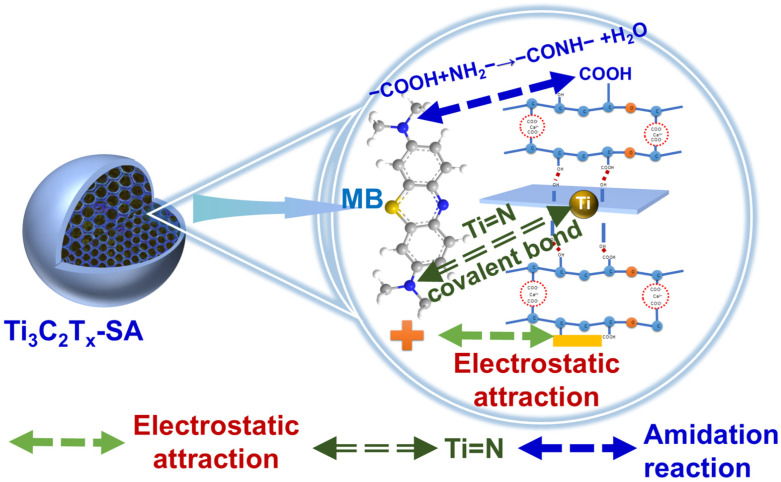
Schematic diagram of adsorption mechanism.

**Table 1 nanomaterials-14-01925-t001:** Obtained fitting parameters of the PFO and PSO kinetic model for the MB adsorption of the 1−Ti_3_C_2_T_x_@SA sample.

C_0_ (mg/L)	q_e exp_ (mg/g)	PFO			PSO		
		q_e cal_ (mg/g)	k_1_	R^2^	q_e cal_ (mg/g)	k_2_	R^2^
100	410.66	407.80	0.1303	0.9906	372.97	6.2596	0.9528
150	633.045	617.12	0.1396	0.9903	572.76	4.4125	0.9568
200	833.5	797.19	0.1706	0.9735	749.15	4.3339	0.9577

**Table 2 nanomaterials-14-01925-t002:** The coefficients of the Weber–Morris model.

C_0_ (mg/L)	k (I)	k (II)	k (III)
100	85.92	46.77	10.48
150	132.10	68.78	10.66
200	156.01	71.51	13.40

**Table 3 nanomaterials-14-01925-t003:** Comparison of adsorption properties between this study and previous reports on the adsorption of MB.

Adsorbent	Initial pH	Equilibrium Time (h)	Maximum Adsorption Capacity (mg/g)	References
Coconut shell biochar	9	32	200	[39]
AgCl-NRs-AC	7	0.5	666	[40]
Graphite powder	7	0.5	476	[41]
Spherical cellulose/carbon	9	6	308	[42]
Agar/graphene oxide aerogel	10	24	578	[43]
Carboxymethyl cellulose/graphene	10	10	222	[44]
Lysine and EDA @graphene aerogel	5.3	24	332	[45]
Ti_3_C_2_T_x_@SA	9	32	969	This work

**Table 4 nanomaterials-14-01925-t004:** Thermodynamic parameters of 1-Ti_3_C_2_T_x_@SA on MB at 298, 308, and 318 K.

∆H (KJ mol^−1^)	∆S (J mol^−1^ K^−1^)	∆G (KJ mol^−1^)		
		298 K	308 K	318 K
23.10	103.01	−7.62	−8.65	−9.68

## Data Availability

Data are contained within the article.

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
