# Peer review of "Enhancing Methylene Blue Adsorption Performance of Ti3C2Tx@Sodium Alginate Foam Through Pore Structure Regulation"

_nanomaterials, 2024, doi:10.3390/nano14231925_

Round 1
Reviewer 1 Report
Comments and Suggestions for Authors
The reported MXene could play as a strong adsorbent of methylene blue (MB) in wastewater. A new family of 2D transition metal carbides/carbonitrides called MXenes found wide applications. MXenes are obtained by selectively etching the A-layers from Mn+1AXn phases and forming Mn+1XnTx (Tx refers to surface functional groups). The authors have discussed synthesizing Mxene and sodium alginate to improve the dye adsorption performance and found it to possess a maximum capacity of 969 mg/g. The synergistic effects of electrostatic adsorption and amidation reaction are explained to be the cause for superior performance.
The submitted work seems to be novel in terms of using Mxene and has some new insights, but the application has been quite well-known for some time. The work in its present form is publishable but needs some revisions before rendering a final decision.
The following points need to be considered.
· Better to avoid the acronym MB in the title.
· Throughout the manuscript, MB has not been explained. The first time they use the acronym MB must be explained in Line 37 (section 1).
· Mxenes are considered to be expensive. why has this material been chosen for adsorption performance?
· What is the role of sodium alignate? Alginic acid can also crosslink with the host material, as stated in doi.org/10.1021/acsaem.1c01111; is this what has been observed in the current work with Mxenes? Please discuss this in detail.
· The authors have claimed honeycomb structure at several places in the manuscript, but there is no clear evidence of being seen in the SEM images. Clarify and resolve it.
· If the author would still frame the honeycomb structure, then the recent literature reported on this porous honeycomb structure (doi.org/10.1002/cplu.202400408) must be well corroborated to benchmark the formation of the honeycomb structure.
· Has Mxene exhibited good recyclability, enabling its repeated use for over 5 cycles? How does this compare with other organic dyes (RhB;doi.org/10.1002/slct.201600867) in terms of their degradation?
· The reactions for complex formation between Ti3C2Tx and MB.
· Lines 396 – 398 how the C-N peak after adsorption confirms MB onto 1-Ti3C2Tx@SA.
· The scale (Binding energy) shown for XPS in Fig. 10 must be reversed. Plotting lower BE to higher BE (left to right).
Reviewer 2 Report
Comments and Suggestions for Authors
The article ‘Enhancing MB Adsorption Performance of Ti3C2Tx@Sodium 2 Alginate Foam Through Pore Structure Regulation’ submitted for review is interesting from a scientific point of view. The search for new sorbents for the removal of dyes (in this case methylene blue) is an ongoing issue. Nevertheless, some additions/corrections are needed before the article can be published.
1. It seems that currently the linear method is no longer suitable for calculating the parameters of the models used (i.e. Pseudo-first-order and Langmuir). A nonlinear method should be used to determine the parameters of all models used. I suggest reading the article http://dx.doi.org/10.1016/j.watres.2017.04.014, which presents in a simple way how to calculate using simple and generally available IT tools such as Excel. In many cases, there are large differences when determining constants from linear and non-linear forms.
2. No information on the number of measurement repetitions. No error bars on figures e.g. Fig.4, 8,9.
3. No detailed information is available on the structure of methylene blue.
4. Introduction - It is beneficial to include the purpose of the paper at the end of the introduction. In the case of the reviewed article, the authors summarised the results of the study rather than formulating the purpose of the study.
5. Section 2.1 - Provide more detailed data on the materials purchased, e.g. purity, country of manufacturer.
6. Line 119 - Add the word ‘spectrophotometer’
7. The adsorption efficiency is denoted in formula 3 as R%. It would be advantageous to denote this efficiency by a different symbol because R2 is denoted as a correlation coefficient (e.g. in Table 1).
8. Figure 6 - It would probably be more accurate to call this model the ‘Weber-Morris’ model.
9. It would be beneficial to provide in a table the coefficients of the Weber-Morris model.
10. Is the value of the maximum adsorption capacity given in Table 2 a measured value or a value calculated, for example, from Langmuir isotherms. This information should be supplemented.
Round 2
Reviewer 1 Report
Comments and Suggestions for Authors
The revised version is suitable for publication as-is, provided the other reviewer/s and the editor have a similar opinion.
Reviewer 2 Report
Comments and Suggestions for Authors No comments.